

# Relationship between isokinetic strength of the knee joint and countermovement jump performance in elite boxers

Chao Chen[1], Zhalel Ali[2], Muhammad Abdul Rehman Rashid[3], Marchibayeva Ulbossyn Samethanovna[2], Guodong Wu[4], Sagidolla Mukhametkali[1] and Tussipkan Dilnur[5]

[1] Shanghai University of Sport, Shanghai, China
[2] Gumilyov Eurasian National University, Astana, Republic of Kazakhstan
[3] Governemnt College University Faisalabad, Faisalabad, Pakistan
[4] Jiangsu Vocational Institute of Commerce, Nanjing, China
[5] National Center for Biotechnology, Astana, Republic of Kazakhstan

Corresponding author
Zhalel Ali, zhalel.ali.phd114@mail.ru

## ABSTRACT

**Background**. The lower limbs play a key role to develop the linear momentum for hitting power in effective boxing. The knee extensor and flexor strength guarantees the dynamic stability of boxers. The insufficient extensor strength of the lower extremities causes compensation during flexion resulting in movement errors or damage to knee joint muscles. This study was conducted to explore the isokinetic concentric strength of the knee flexor and extensor and the relationship between isokinetic knee extensors strength and countermovement jump (CMJ) performance in elite boxers.

**Methods**. Thirteen elite male boxers (Age: $25.15 \pm 3.98$ years, height $1.72 \pm 0.04$ m, weight $61.82 \pm 10.46$ kg, training years $= 11.56 \pm 2.67$ years) performed the CMJ, and the isokinetic knee test was performed using the Biodex dynamometer.

**Results**. The maximal isokinetic peak torque of the knee extensor and flexor muscles was recorded at three angular velocities ($60°/s$, $180°/s$, and $240°/s$) on both sides of the legs. The relative peak value of torque in the knee extensors decreased significantly with increasing angular velocity. A difference in relative peak torque (RPT) was only seen at $60°/s$ in knee flexors. However, the H/Q ratio increased as the velocity increased from $60°/s$ to $240°/s$ ($P < 0.05$). The highest peak torque was found in the knee extensors at a velocity of $240°/s$ ($r = 0.73$, $P < 0.001$). The correlation between RPT and vertical jump height was the strongest at $240°/s$. The strongest relationship was found between the height of the CMJ and the RPT of the deficit of knee extensors.

**Conclusions**. We suggest that explosive force training of the isokinetic muscles should be optimally carried out at a speed of $240°/s$. The results of this study provide a reference for boxers to improve their jump height and lower-limb explosive strength through isokinetic strength training of the knee flexor and extensor.

## INTRODUCTION

During boxing competitions, referees score boxers on a 10-point scale, which has shifted the focus from simply the number of punches thrown to quality punches (punch force) landed on the target area, superiority (control of the opponent), competitiveness (positive offence), and so on. The most crucial factor in effective boxing is hitting power, which is determined bases on the comprehensive strength and the offensive manifestation of this power (*Pierce Jr et al., 2006*; *Smith, 2006*; *Chaabène et al., 2015*). A boxer's punching action originates from the force applied to the ground, the rotation of the hips and torso, and finally extending the arm to hit the opponent (*Pierce Jr et al., 2006*; *Chaabène et al., 2015*). The lower limbs play a key role in this process through the linear momentum of the transmission force (*Smith, 2006*). It has been reported that the pedal movement of the lower extremities is responsible for the production of strike power (*Cheraghi et al., 2014*), which is the primary contributor to effective attacks. In high-level amateur boxing, well-developed muscle strength of the upper and lower limbs is necessary. Numerous studies have highlighted the importance of lower-limb strength on maximal punch force in well-trained boxers (*Loturco et al., 2016*; *Dunn et al., 2022*). *Dunn et al. (2022)* indicated that lower-limb strength measurements in highly trained boxers were able to discriminate between the most and least powerful punchers (*Dunn et al., 2022*). The muscle strength of the lower extremities is the boxer's primary source of strength during impact, and the extension of the knee joint is the key link to lower limb pedal transfer force. The strength level of the knee flexor and extensor has a crucial influence on the impact force of the punch, and therefore influences competitive performance of boxers (*Loturco et al., 2016*; *Zhou et al., 2022*).

Explosive force is defined as the capacity to exert the maximum amount of force in the shortest possible time (*Piorkowski, Lees & Barton, 2011*; *Walilko et al, 2005*). This is a crucial component of boxing, and a high level of strength and speed is necessary to enhance explosive power (*Smith, 2006*; *Walilko et al, 2005*). Due to the grading on a 10-point scale, effective boxing focuses on a combination of strength and speed, and a highly developed explosive force can effectively enhance the impact of punches (*Kawamori & Haff, 2004*; *Pierce Jr et al., 2006*; *Smith, 2006*). Rapid expansion and compound training are the primary strength training methods used by boxers (*Martsiv, 2014*), while the vertical longitudinal jump is a commonly used test to measure explosive force in the lower extremities (*Wisløff et al., 2004*). Extensor muscle strength is significantly correlated with the height of the longitudinal jump (*Fischer et al., 2017*) and special combat techniques (*Loturco et al., 2014*; *Zaggelidis & Lazaridis, 2013*; *Zangelidis et al., 2012*), making it an effective indicator of the boxer's lower limb explosive force and overall boxing ability.

The strength of the knee flexor and extensor is essential to guarantee the integrity and stability of boxers during matches. Research has shown that insufficient extensor strength of the lower extremities causes compensation during flexion, which may result in movement errors or injuries to the flexion knee joint muscles (*Dauty et al., 2014*; *Kellis & Kouvelioti, 2009*). The knee joint extensor muscles play a significant role in the landing of athletes and

work in tandem with the flexible muscles of the hip joint to ensure stability of the body (*Yeow, Lee & Goh, 2011*).

The knee joint is particularly important in maintaining high stability, and it has been demonstrated that the extensor muscles and flexion knee joint muscles play different roles in technical movements. The H/Q value is defined as the ratio of the flexion muscles to the extensor muscles, which reflects the muscle strength balance between the antagonistic and active muscles in joint activity. It is an indicator of joint stability, as well as the cooperative force ability of the technical action and the risk of injury. A ratio beyond the normal range indicates that the tensile stress of the weak muscle group and ligament is high, which may lead to muscle and ligament injuries (*Daneshjoo et al., 2013*; *Zvijac et al., 2014*). Some studies have also shown that when an athlete's joint muscle strength ratio (extensor *vs.* flexor) is maintained close to 1, it plays a positive role in preventing the occurrence of injury and improves coordinated actions. Together, they provide a reasonable and effective execution of physical actions. Furthermore, the ratio of bilateral muscles on the opposite side refers to the joint flexion group ratio and extensor group peak value between the left and right sides, which is an important index of bilateral muscle balance. It has been reported that the ratio of homonymous muscles plays a positive role in preventing the occurrence of injury and the coordinated development of action (*Dauty et al., 2014*; *Skou et al., 2016*; *Qingguang et al., 2016*).

Previous studies have confirmed an association between isokinetic knee joint measurements and jump performance (*Lenetsky, Brughelli & Harris, 2015*). However, few studies have investigated the isometric and isokinetic muscle strength of various parts of a boxers' body, such as the legs, trunk, shoulders, and elbows (*Tasiopoulos et al., 2018*; *Zhou et al., 2022*). A previous study conducted on amateur boxers examined the isokinetic muscle strength of the rotators of the glenohumeral joint, bilateral, unilateral, and functional ratios (eccentric peak torque: concentric peak torque) to understand the variation in these muscle strength characteristics according to performance level (*Fischer et al., 2017*). However, to our knowledge, no study has yet elucidated the relationship between isokinetic knee flexor and extensor strength and vertical jumping in elite boxing. Therefore, the aim of this study was to investigate the relationship between isokinetic knee muscle strength and vertical jump performance. We hypothesized that a correlation would exist between the extensor muscle strength of the knee joint and vertical jump height. We further hypothesized that the relationship between the peak torque of the extensor and the vertical take-off height is the highest at a certain angular velocity.

## MATERIALS & METHODS

### Participants

Thirteen elite male boxers (25.15 ± 3.98 years; mean height 1.72 ± 0.04 m; mean weight 61.82 ± 10.46 kg; training years 11.56 ± 2.67 years) volunteered to participate in the present study. All subjects were trained daily (3–5 h per day) and routinely completed training at national and international levels. Three boxers were left-handed. The dominant leg was defined as the rear leg, and the non-dominant leg was defined as the lead leg
when in the fighting stance. All participants self-reported dominant and non-dominant legs. For all subjects, the right (rear) leg was the dominant leg in this study. They were healthy and had no injuries six weeks prior to the test. The amateur boxers signed the informed consent form prior to participation and volunteered to participate in this study, which was approved by the Shanghai University of Sport Research Ethics Committee (approval number: 102772021RT031) and was conducted in accordance with the Helsinki declaration.

## Design and procedures
### Isokinetic strength assessment

The isokinetic muscle strength of the lower extremities was assessed using the Biodex dynamometer (Biodex Medical Systems IV; Shirley, NY, USA). Before the test, participants all completed a 15-minute warm-up protocol, comprising 10 min of stationary cycling and 5 min of dynamic stretching. Subsequently, participants performed a 1-minute isokinetic muscle force adaptation exercise, followed by a 1-minute rest period before the start of the formal test. Participants were placed in the seated position, with their legs raised and their knees and buttocks firmly secured at 90o. Knee extension was limited to 0o and the knee angle was restricted to 110o. Limb weight was corrected for the effect of gravity and aligned with the motion axis of the knee joint. A belt was used to fix the thighs, buttocks, and chest, ensuring that only the knee joint was measured. The participants' legs were positioned at 90° and a device was used to extend the knee joint range of motion from 90° to 180°. The tests were carried out five trials at angular velocities of 60°/s, 180°/s, and 240°/s, with a 1-minute rest period between each trial, of which the highest value was used for further analyses. The left and right leg tests were separated by a 3-minute rest period. The dominant leg was always tested first, followed by the non-dominant leg. During these tests, verbal encouragement was provided to motivate the participants to perform at their maximum strength and speed (*Bamaç et al., 2008*). The following parameters were selected for quantitative analysis: relative peak torque (RPT), hamstring/quadriceps (H/Q) ratio, and bilateral ratios. The best peak torque of the knee extensor and flexor muscles was recorded at three angular velocities (Nm). RPT was used as the peak torque in relation to the body mass (Nm/kg). The conventional H/Q ratio was calculated using the absolute peak torque of the extensor and flexor muscles of the knee joint to balance the relationship between the extension and flexion of the knee joint. Bilateral ratios were calculated as the RPT of dominant leg divided by the RPT of non-dominant leg.

### Countermovement jump (CMJ) testing

A Smart Jump (Fusion Sport, Coopers Plains, Australia) was applied to measure the explosive force of the lower limbs through CMJ testing. The Smart Jump showed high reliability of the countermovement jump test (ICC = 0.94) in a prior study (*Loturco et al., 2016*). Jump height was estimated from flight time using the formula: Jump Height = 9.81 * (flight time)2/8. Prior to the test, participants completed a warm-up consisting of 3 min of running on a treadmill at 5 km/h, followed by 5 min at 8 km/h. Participants were familiarized with the testing process before performing five CMJ attempts with a 30-second rest between each attempt (*Harrison et al., 2013*; *Wilhelm et al., 2013*). Boxers in

the CMJ were instructed to do a downward movement followed by complete leg extension and were permitted to determine the countermovement amplitude to avoid changes in jumping coordination (*Bosco, Mognoni & Luhtanen, 1983*). A sufficient amount of rest was provided between trials and the best of three trials for each jump condition was recorded. The highest value of the three trials was used for later analysis.

## Statistical analyses

Statistical analyses were performed using Statistical Software 10 (StatSoft, Tulsa, OK, USA). The normality of each variable was tested using the Shapiro–Wilk test. As data displayed normal distribution, a two-way analysis of variance (ANOVA) with repeated measures [2 (leg: dominant, non-dominant) $\times 3$ (velocity: 90°/s, 180°/s, and 240°/s)] was conducted to analyze the relative knee flexors/knee extensors PT. For the H/Q ratios, a two-way ANOVA with repeated measures [2 (leg: dominant, non-dominant) $\times 3$ (velocity)] was used. Post hoc analyses were performed with the Bonferroni test. The effect size (ES) of ANOVA was evaluated using eta squared ($\eta^2$) and interpreted as follows: small $= \eta_p^2 \geq 0.01$ and $< 0.06$; medium $= \eta_p^2 \geq 0.06$ and $< 0.14$; large $= \eta_p^2 \geq 0.14$. Pearson product-moment correlation coefficient (r) and linear regression analysis were applied to assess the relationship between the absolute or relative PT and vertical jump, with absolute or relative PT as the dependent variable and vertical jump as the independent variable. The following thresholds were used to qualitatively assess correlations: $< 0.1 =$ trivial; 0.1 to 0.3 $=$ small; 0.31 to 0.5 $=$ moderate; 0.51 to 0.7 $=$ large; 0.71 to 0.9 $=$ very large; $> 0.91 =$ nearly perfect, based on *Hopkins (2002)*. This analysis was performed only for the knee extensors. For each muscle group, the PT of each leg was summed to assess the relationship. Statistical significance was set at $P < 0.05$.

## RESULTS

### Absolute peak torque

The relative PT of all boxers can be seen in Table 1. For the peak torque of knee extensors, no significant leg $\times$ angular velocity interactions ($F = 1.123$, $p = 0.316$, $\eta p^2 = 0.045$) were observed, but there was a significant main effect for angular velocity ($F = 249.924$, $p < 0.001$, $\eta p^2 = 0.912$). For the peak torque of knee flexors, there was no significant leg $\times$ angular velocity interaction ($F = 2.331$, $p = 0.108$, $\eta p^2 = 0.089$) or main effect for leg, but there was a main effect for angular velocity ($F = 25.133$, $p < 0.001$, $\eta p^2 = 0.512$). Bonferroni analysis revealed a significant decrease in relative PT value in the knee flexors as the angular velocity increased from 180°/s to 240°/s. Bonferroni post hoc test revealed a significant decrease in relative PT of the knee extensors as the test velocity increased from 60°/s to 240°/s in dominant leg ($p < 0.001$) and non- dominant leg ($p < 0.001$), while for the knee flexors, this trend was only observed between 60°/s and 180°/s in dominant leg ($p = 0.003$) and non-dominant leg ($p < 0.001$). These results indicate that the effects were independent of the angular velocities.

### H/Q ratio

There was no significant group $\times$ angular velocity interactions for the H/Q ratio ($F = 2.836$, $p = 0.082$, $\eta_p^2 = 0.106$), but there was a main effect for angular velocity ($F = 121.239$,

**Table 1  Relative peak torque in the dominant and non-dominant leg at three angular velocities.**

| Velocity | Knee extensors | | Knee flexors | |
|---|---|---|---|---|
| | D | ND | D | ND |
| 60°/s (Nm/kg) | 2.66 ± 0.41 | 2.50 ± 0.42 | 1.42 ± 0.22 | 1.43 ± 0.22 |
| 180°/s (Nm/kg) | 1.90 ± 0.24[a] | 1.84 ± 0.26[b] | 1.26 ± 0.16[a] | 1.21 ± 0.19[a] |
| 240°/s (Nm/kg) | 1.56 ± 0.25[a] | 1.60 ± 0.32[a] | 1.30 ± 0.16 | 1.22 ± 0.24 |
| CMJ (cm) | 37.67 ± 4.51 | | | |

**Notes.**

D, dominant leg; ND, non-dominant leg; CMJ, Countermovement jump.
Significant difference between pairs of consecutive angular velocities.
[a] ($P < 0.01$).
[b] ($P < 0.001$).
Data are presented as mean ± standard deviation.

$p < 0.001$, $\eta_p^2 = 0.835$). The *post hoc* Bonferroni test revealed that the H/Q ratios significantly increased ($p < 0.001$) as the velocity increased from 60°/s to 240°/s, regardless of the leg (Table 2).

### Bilateral ratios

No significant differences in the bilateral ratios were observed at 60°/s, 180°/s, and 240°/s ($P > 0.05$) (Table 3).

### Relationships between the countermovement jump and relative peak torque

The CMJ performance is presented in Table 1. Linear regressions and correlation coefficients were computed to examine the relationship between CMJ height and relative PT of the knee extensors at the three angular velocities. The results showed significant ($p < 0.05$) positive correlations at high velocities (180°/s and 240°/s) (Table 4). The dominant leg's knee extensors at a velocity of 240°/s had the strongest correlation with CMJ height among the relative PT (Table 4). There is a significant correlation ($p < 0.05$) between the height of the CMJ and the relative PT of the dominant and non-dominant knee extensor deficits only at 240°/s. Linear regression results showed that the peak torque of knee extensors at 60°/s in the dominant leg ($R^2 = 0.119$, $F = 1.490$, $p = 0.248$) and non-dominant leg ($R^2 = 0.284$, $F = 4.360$, $p = 0.061$) explained 11.9% and 28.4% of CMJ height, respectively. At 180°/s, the peak torque of knee extensors in the dominant leg ($R^2 = 0.325$, $F = 5.287$, $p = 0.042$) and non-dominant leg ($R^2 = 0.336$, $F = 5.568$, $p = 0.038$) explained 32.5% and 33.6% of CMJ height, respectively. The peak torque of knee extensors at 240°/s in the dominant leg ($R^2 = 0.428$, $F = 8.217$, $p = 0.015$) accounted for 42.8% of CMJ height. The peak torque of knee extensors at 240°/s in the non-dominant leg ($R^2 = 0.376$, $F = 6.635$, $p = 0.026$) accounted for 37.6% of CMJ height.

### DISCUSSION

This study investigated the isokinetic muscle strength of the knee flexor and extensor and the relationship between knee extensor strength and CMJ performance in elite boxing athletes. Overall, we found no significant angular differences between the dominant and

**Table 2  Hamstring to quadriceps ratio (H/Q) according to leg and angular velocity variables.**

| Velocity | D | ND |
|---|---|---|
| 60°/s | 0.53 ± 0.02 | 0.58 ± 0.07 |
| 180°/s | 0.67 ± 0.07[a] | 0.66 ± 0.09[a] |
| 240°/s | 0.77 ± 0.10[a] | 0.84 ± 0.11[a] |

**Notes.**

Significant difference between pairs of consecutive angular velocities.

[a] ($P < 0.001$).

D,  dominant leg;  ND,  non-dominant leg.

Data are presented as mean ± standard deviation.

non-dominant legs at velocities of 60°/s and 240°/s. However, the H/Q ratio increased as the velocity increased from 60°/s to 240° /s. The highest PT for the knee extensor was observed at a velocity of 240°/s. Additionally, no significant positive correlations were found between the homonymous muscle groups on opposite sides of the knee joint at maximum and fast strength.

Boxing is a sport that involves symmetrical technical movements, including defensive and offensive actions ("hit out" and "get back"), which are repeatedly performed in a dynamic competition. As a result, the muscles involved experience alternating centripetal and centrifugal forces, with a closer ratio of knee joint muscle flexion and extension indicating better muscle balance, approaching a ratio of 1. A ratio of knee joint flexion and extension that ranges from 50% to 80% for the flexor and extensor groups is considered less risky (*Croce et al., 1996*). However, this ratio varies among athletes of different sports. For example, the knee joint flexion and extension ratio was 60%–69% in slow-speed tests (60o/s) (*García-Ramos et al., 2017*), 70%–79% in medium-speed tests (180o/s), and 80%–95% in rapid tests (300°/s) (*Fischer et al., 2017*). In the present study, the H/Q ratio in Chinese boxers was found to be between 54% and 85% at 60°/s, 180° /s, and 240°/s. The research results of this study are basically consistent with previously published relevant research results. *Zhou et al. (2022)* found the H/Q ratios are typically between 55% and 80% at 60°/s, 180°/s and 240°/s in amateur boxers. In conclusion, the H/Q ratio has shown an increase trend with increasing angular velocity in different studies. We observed that this is caused by a major decrease of strength brought on by knee extensors strength. However, when the angular velocity increased, there was not a significant decrease in the strength generated by knee flexors strength. This is likely because of the fact that hamstring muscles have faster muscle fibers than quadriceps muscles (*Garrett Jr, Califf & Bassett 3rd, 1984*). Although there is currently no established range for the H/Q ratio in boxers, a greater difference in knee joint flexion and extension muscle force indicates poorer joint stability and a higher risk of injury. It has been observed that the normal range of H/Q ratio is more than 0.6, and that if it is less than 0.6, the imbalance between flexion and extension reduces stability and increases the risk of injury 17 times (*Zvijac et al., 2014*). This holds true not only for athletes, but also for non-sporting individuals, and it is essential to ensure strength balance between the hamstring and quadriceps.

The heterolateral knee joint muscle is a known risk factor for muscle injury (*Daneshjoo et al., 2013*). Based on previous studies, muscle balance differences can be divided into

**Table 3  Dominant and non-dominant leg ratio of flexor and extensor strength across angular velocities.**

| Velocity | Knee extensors | Knee flexors |
| --- | --- | --- |
| 60°/s | 1.08 ± 0.14 | 1.00 ± 0.13 |
| 180°/s | 1.05 ± 0.06 | 1.03 ± 0.06 |
| 240°/s | 1.08 ± 0.13 | 1.00 ± 0.11 |

Notes.
Data are presented as the mean ± standard deviation.

**Table 4  Relationships between the height of the countermovement jump and relative peak torque of the dominant and non-dominant knee extensors across angular velocities.**

| | r | p-value |
| --- | --- | --- |
| RPTextensorD60 | 0.345 | 0.248 |
| RPTextensorND60 | 0.533 | 0.061 |
| RPTextensorD180 | 0.570 | 0.042 |
| RPTextensorND180 | 0.580 | 0.038 |
| RPTextensorD240 | 0.654 | 0.015 |
| RPTextensorND240 | 0.613 | 0.026 |

Notes.
Statistically significant associations $P < 0.05$.
RPT, relative peak torque; D, dominant; ND, non-dominant.

three levels: (1) a difference of less than 10% (normal), (2) a difference of 10%–20% (risk), and (3) a difference of more than 20% (abnormal) (*Croisier et al., 2008*; *Daneshjoo et al., 2013*). The current study suggests that a difference of approximately 10% should ensure muscle balance. Overall, we found no significant differences in the maximum strength, fast strength, or strength endurance in the heterolateral knee joint muscles of Chinese elite boxers. Additionally, the difference in the ratio of bilateral muscles on the other side was found to be less than 10%, indicating that the heterolateral flexor and extensor strengths of the knee joints in these athletes were well balanced. These findings are in agreement with a previous study conducted by *Pierce Jr et al. (2006)*, and suggest that elite Chinese boxers have relatively consistent ratios of bilateral muscles on both sides of their knee joints. It is worth noting that heterolateral muscle strength differences will increase as the training experience does (*Lisowska, Murawa & Ogurkowska, 2020*). Elite boxers' greater physical activity and longer duration of exercise may increase their risk of injury. It has been seen that repeating the same patterns of motion over a period of years might have negative effects, which often exhibit as contralateral muscle strength imbalances (*Tourny-Chollet, Seifert & Chollet, 2009*). Therefore, strength and conditioning coaches should regularly measure and correct heterolateral knee extensor and flexor strength differences to prevent injuries.

Overall, a relationship was observed between the isokinetic strength of the knee extensors and CMJ height. Significant correlations were found between knee extensor strength and vertical jump height at angular velocities of 180°/s and 240°/s. However, no significant correlation was observed at 60°/s, leading to the conclusion that the angular velocity of the PT is a critical factor influencing this relationship. The relationship between knee flexor

strength and vertical jump performance was not evaluated in the study. This is because of the insignificant contribution of the knee joint flexors in the vertical jump test, as pointed out in *Struzik & Pietraszewski (2019)*. *Diker et al. (2022)* indicated the important role of knee joint extensors strength in sprint performance. These studies exhibited that knee joint extensor strength plays an important role in jump and sprint performance. In the present study, the correlation coefficient between the RPT and vertical jump height was found to be the highest at 240°/s. Even at 180° /s, there was still a correlation, although the correlation coefficient lower. However, regression results indicated a significant relationship between CMJ height and the RPT, which was found only at 240°/s in the dominant and non-dominant legs. Based on these results, we suggest that explosive force training of the isokinetic muscles should be conducted at a velocity of 240°/s.

The vertical jump is a movement primarily dominated by extensor function (*Malliou et al., 2003*). In CMJ exercises, the contribution rates of the knee joint, buttocks, and ankle joint were approximately 49%, 28%, and 23%, respectively (*Hubley & Wells, 1983*). It has been noted that the range of motion of the knee joint is greater than that of the hip joint (*Bobbert et al, 1996*). Furthermore, the extensor muscles and tendons of the knee joint possess high elasticity, which allows them to be retracted through elasticity (*Kubo, Kawakami & Fukunaga, 1999*). The stiffness of the vertical hip is considered an effective indicator of competitive activity in boxers (*Davis et al., 2015*; *Davis et al., 2018*). The present research indicates that the knee extensor strength of boxers at 180°/s and 240°/s was proportional to the vertical takeoff height, thereby suggesting that knee extensor strength and explosive force ability play an increasingly crucial role in boxing matches. The present study also had limitations, which require further discussion. First, we investigated elite male athletes, which limits the ability to generalize the findings toward other-level boxers. Further studies should engage female boxers and other-level boxers. Second, we did not use a force platform, which might have resulted in more findings in assessing strength and power in vertical jump tests.

## CONCLUSIONS

To conclude, the H/Q values of the knee joint in both the dominant and non-dominant legs of Chinese boxers were observed to increase with an increase in test speed. This finding suggests that the flexion muscle of the knee joint in Chinese boxers is relatively weak, thereby increasing the risk of knee joint injury. The strength of the extensor muscle, however, was found to be balanced, with a relatively coordinated strength ratio of the homonymous muscles on the other side. Our study also found a significant correlation between knee extensor strength and vertical jump height, particularly at angular velocities of 180°/s or 240°/s. Notably, a velocity of 240°/s showed the strongest correlation, indicating that muscle explosive strength training should be performed at this speed. Conducting strength training at higher speeds could potentially enhance lower-limb explosive strength, which may contribute to improving punch performance.

### Funding

The research was funded by Shanghai Key Lab of Human Performance of Shanghai University of sport, China, Project ID: NO. 11DZ2261100. The funders had no role in study design, data collection and analysis, decision to publish, or preparation of the manuscript.

### Grant Disclosures

The following grant information was disclosed by the authors:
Shanghai Key Lab of Human Performance of Shanghai University of sport, China: 11DZ2261100.

### Competing Interests

The authors declare there are no competing interests.

### Author Contributions

- Chao Chen conceived and designed the experiments, performed the experiments, analyzed the data, authored or reviewed drafts of the article, and approved the final draft.
- Zhalel Ali conceived and designed the experiments, performed the experiments, analyzed the data, authored or reviewed drafts of the article, and approved the final draft.
- Muhammad Abdul Rehman Rashid conceived and designed the experiments, performed the experiments, analyzed the data, prepared figures and/or tables, and approved the final draft.
- Ulbossyn Marchibayeva performed the experiments, prepared figures and/or tables, authored or reviewed drafts of the article, and approved the final draft.
- Guodong Wu conceived and designed the experiments, performed the experiments, analyzed the data, authored or reviewed drafts of the article, and approved the final draft.
- Sagidolla Mukhametkali analyzed the data, prepared figures and/or tables, and approved the final draft.
- Tussipkan Dilnur performed the experiments, prepared figures and/or tables, and approved the final draft.

### Human Ethics

The following information was supplied relating to ethical approvals (i.e., approving body and any reference numbers):

The amateur boxers volunteered to participate in this study, which was approved by the Shanghai University of Sport Research Ethics Committee (approval number: 102772021RT031) and in accordance with the Helsinki declaration.

### Data Availability

The raw measurements are available in the Supplementary File.

## Supplemental Information

Supplemental information for this article can be found online at http://dx.doi.org/10.7717/peerj.16521#supplemental-information.

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
