# Peer review of "Relationship between isokinetic strength of the knee joint and countermovement jump performance in elite boxers"

_PeerJ, doi:10.7717/peerj.16521_

## Round 0.1 · original submission · Major Revisions

Following review there are a number of concerns raised by the reviewers that I consider requires major revisions. Please specifically focus on addressing the following:

1. Appropriate terminology (e.g. torque) and nomenclature.

2. A balanced and comprehensive discussion reflective of the size of effects / correlations observed.

3. Including details about the methods and statistical analysis

Please note that all reviewer comments should be addressed and a full response provided for each point raised.

Reviewer 1 ·

Basic reporting

The literature is, in its majority, relevant and well referenced. The structure of the manuscript conforms to PeerJ standards and the raw data are provided. However, there are cases that the terminology used is not optimal and in some cases is not uniform throughout the text.

Experimental design

The rationale of the study is developed and stated sufficiently. It is stated how the research fills an identified knowledge gap. It seems that technical and ethical standards were applied. However, concerning the Materials and Methods, additional details are required for the replication of the experimental design.

Validity of the findings

Not all findings are reported sufficiently. Conclusions are well stated, but not entirely linked to the research question and not based on the findings of the study per se.

Additional comments

Abstract
• L43: Define D and ND at L36.
• L43-45: It is recommended to move this part of the text in the “Conclusions” section of the Abstract.

Introduction
• L53: Please, provide more details about the “other factors”.
• L59: It is recommended to split the sentence here.
• L61: “excellent strength quality”: Please, elaborate on the criterion for such a statement.
• L66: This statement can be benefited by adding a reference(s).
• L69: “damage”: Do you mean injury?
• L76-90: It is recommended to move this part of the text after L66.
• L82: Provide details for the “super isometric training”.
• L82: “innervations”: Is this terminology correct?
• L85: It is recommended to delete “longitudinal”. Also at L87 and elsewhere.
• L86: It is recommended to use “test” instead of “index”.
• L94: The Malliou et al. (2003) study is about soccer players. It is suggested to revise this sentence using references related to boxing.
• L96: It is recommended to provide a definition for the functional ratios.
• L98: “constant-speed” = isokinetic. Note that angular velocity rather than speed is a more commonly used term for the isokinetic tests.
• L99: Please, insert the Aim of the study is missing.
• L101: “isokinetic strength” = isokinetic torque.
• L103-105: It is recommended to move this part of the text at L99.
• L103: Although obvious, define CMJ.

Materials & Methods
• L109: At L103, it is stated that 20 elite Chinese boxers were analyzed. In this section, and also in the attached dataset, the sample size seems to be 13. Please, clarify this.
• L109: State how many boxers were (if) left-handed.
• L111: …at national…
• L115: …and was conducted in accordance …
• L123: Please provide the rational for using a supine rather than the sitting position for the isokinetic tests.
• L125: It is recommended to use “trials” instead of “times”.
• L125-127. It is recommended to move this part of the text to L130.
• L132-140: Thank you for providing in detail the examined parameters. However: define all abbreviations (PT, BW, etc.); use body mass than body weight; note that PT/BW is already defined in L133; you mention two angular velocities, but three were tested (please clarify at L136); please, explain how the dominant and non-dominant legs were defined; were the measurements of the dominant and non-dominant leg executed in a random order? It is also recommended to use symmetry index to report the findings.
• L141-150: For better clarity, it is suggested to provide how the CMJ height was calculated. In addition, state in brief which criteria were applied to consider as valid the “consistent jumping coordination pattern”.
• L151-167: The Statistical Analyses subsection needs more clarifications. At first, I suggest that you remove the Roy et al. (1984) reference from the eta squared thresholds. In addition, in the repeated measures two-way ANOVA, the factor “leg” is comprised of the lead and the rear leg, but in the rest of the manuscript the factor seems to be about the dominant and the non-dominant leg. This has to be clarified. At L153, it is stated that “For normally distributed data”. Thus, were there non-normally distributed data and if so, what was the statistical analysis in that case? Another topic for clarification is why the eta squared rather than the eta partial squared was used to define the effect size? Furthermore, it has to be clear what data were used for the statistical analysis: the average values of the 3 (CMJ)/5 (isokinetic tests) trials of the maximum recorded within the trials? The need for clarification is also evident on the rationale to conduct the regression analysis only for the knee extensors and not also for the flexors. Finally, was there any estimation that the sample size was adequate to establish meaningful effect sizes for the selected statistical analyses?

Results
• L169-214: Thank you for providing the Results. However, some parts of this part of the text seem appropriate for the Introduction or the Methods sections (L183-190 & L198-203). Furthermore, no details are provided for the results of the jump height in the CMJ and the statistics of the regression analysis. In addition, state F, p, and eta partial squared for all presented results.
• L174: “P=0.000” should be written as “P<0.001” (also at L175).
• L206: Insert a separate subsection for the results of the correlation and regression analyses.

Discussion
• L219: The inter-limb difference is now presented as “front and rear leg”. It is recommended to use the same terms throughout the text.
• L237: It is recommended to clarify “two sides of the joint”: For some readers, it can be interpreted as the distal and proximal part surrounding the joint, i.e., the thigh and the shank are the two sides of the knee joint.
• L240: It is recommended to use another term for “suspicious”.
• L256: It is recommended to use “was lower” than “decreased”.
• L263: …respectively (Hubley…
• L272: If there is a weakness in the Discussion, it is the absence of the limitations of the study and the lack of suggestions for future research.

Conclusions
• L278: Please, clarify if you mean jump height when referring to “vertical take-off height”.
• L281-282: Please clarify “isokinetic muscles”.

Tables
• T2-L4: Please, clarify if you mean between the dominant and non-dominant leg.
• T3: Please, depict all data within Table 3 with 2 decimals.
• T4-L2: In the text, RPT was defined as PT/BW. Please, use the same abbreviation throughout the text.
• T4: Please, depict all p values within Table 4 with 3 decimals.
• T5-L2: In the text, RPT was defined as PT/BW. Please, use the same abbreviation throughout the text.
• T5: Please, depict all p values within Table 5 with 3 decimals.

Annotated reviews are not available for download in order to protect the identity of reviewers who chose to remain anonymous.

Reviewer 2 ·

Basic reporting

- The article contains significant conceptual errors. There is no such thing as knee joint strength, how can a joint generate force?

- It is necessary to use the correct biomechanical nomenclature. The manuscript does not refer to strength, but to torque - it is a major conceptual error. It's also not about constant-speed, but about angular velocity.

- The review of the literature is quite superficial. Several key works on this problem are missing.
For example: Diker et al. 2022 or Struzik and Pietraszewski 2019.

- There are no units for reported values, e.g. in tables.

Experimental design

- The lack of measurements of jump variables using a force plate is a very big limitation. Please note that the contact mat has a much larger measurement error and lower accuracy.

Validity of the findings

- The conclusions are very bold. Looking at the r values obtained, the conclusions should be much more cautious.

Additional comments

The article should be prepared with much greater care, especially conceptual. Many limitations are also not highlighted.
The discussion is very poor and short. More attention should be paid to the cause of such results. There should be more comparisons to the work of other authors.

---

## Round 0.2 · Minor Revisions

Thank you for your updated manuscript and responses. As you can see, reviewer 1 has some further comments which I would like you to address. Please provide a responses document along with your revised manuscript upon resubmission.

Reviewer 1 ·

Basic reporting

In the resubmitted paper, the author(s) did an exceptional work to address the concerns raised in the initial round of reviewing. There as still some issues regarding the bibliographic evidence and the terminology used.

Experimental design

In the resubmitted paper, the author(s) addressed most of the topics raised in the initial round of review. However, there are still a number of topics that need to be addressed, as pointed out in the additional comments.

Validity of the findings

There is space for improvement regarding the statistical analyses, as pointed out in the respecitive additional comments.

Additional comments

Specific comments:

Abstract
• L38: Rephrase for better clarity, especially for the “knee joint strength” and the “integrity”.
• L58: Again, clarification is needed: the study examined the relationship of jump height and isokinetic torque, not the relationship between the vertical ground reaction forces and isokinetic torque. Please rephrase both points of the abstract.

Introduction
• L65-66: Although more performance factors are now mentioned, further elaboration on “quality” and “superiority” is needed in terms of their quantitative assessment.
• L61: “excellent strength quality”: Please, elaborate on the criterion for such a statement.
• L74, L96: It is recommended to split paragraphs at these points.
• L111: Provide a reference for the inserted statement.
• L114: Use another term for “homonymous”. The same at L300.

Materials & Methods
• L137-140: As no measurements were conducted for the establishment of leg dominancy, it is recommended to use the technique terms: front and rear leg rather than non-dominant and dominant leg, respectively.
• L159-160: How was a possible order effect controlled as no randomization between tested legs was applied in the testing procedure?
• L163: Report the bilateral ratio used.
• L125-127. It is recommended to move this part of the text to L130.
• L169-178: There is still no clear information if the CMJ height was calculated based on flight time or vertical impulse/take-off velocity. It is recommended to include the references mentioned in the Author(s)’ reply. Finally, state the reliability of the test.
• L178. …further analysis.
• L179-195: It is still stated “For normally distributed data”. Again, define if there were non-normally distributed data. In this case, what was the statistical analysis? It is recommended to include in the text the rationale developed in the Author(s)’ reply, together with the mentioned references.

Results
• It seems that the results from the CMJ do not fit well within Table 1. It is recommended to mention the results for the CMJ in the text.
• There is no depiction of the bilateral ratio in Tables 1 and 2.

Discussion
• L258: Cite the “previous studies”.
• L291: Elaborate on the limitations of the study.

References:
• L317: Provide the full journal title.
• L412-414: Zangelidis, G., Lazaridis, S. N., Mavrovouniotis, F. & Malkogeorgos, A. (2012). Differences in vertical jumping performance between untrained males and advanced Greek judokas. Archives of Budo, 8(2), 87-90. doi:10.12659/AOB.882775

Reviewer 2 ·

Basic reporting

No comment.

Experimental design

No comment.

Validity of the findings

No comment.

Additional comments

After the corrections made by the authors, the manuscript seems, in my opinion, appropriate.

---

## Round 0.3 · accepted · Accept

Thank you for considering all of the reviewer comments. I am satisfied that you have addressed all concerns and am pleased to accept this manuscript.